ecology

demography, population trends, migration, conservation, productivity

# Covariation in population trends and demography reveals targets for conservation action

Catriona A. Morrison[1], Simon J. Butler[1], Robert A. Robinson[2], Jacquie A. Clark[2], Juan Arizaga[3], Ainars Aunins[4,5], Oriol Baltà[6], Jaroslav Cepák[7], Tomasz Chodkiewicz[8,9], Virginia Escandell[10], Ruud P. B. Foppen[11,12], Richard D. Gregory[13], Magne Husby[14,15], Frédéric Jiguet[16], John Atle Kålås[17], Aleksi Lehikoinen[18], Åke Lindström[19], Charlotte M. Moshøj[20], Károly Nagy[21], Arantza Leal Nebot[22], Markus Piha[23], Jiří Reif[24,25,26], Thomas Sattler[27], Jana Škorpilová[28], Tibor Szép[29], Norbert Teufelbauer[30], Kasper Thorup[31], Chris van Turnhout[11,12], Thomas Wenninger[32] and Jennifer A. Gill[1]

[1]School of Biological Sciences, University of East Anglia, Norwich Research Park, Norwich NR4 7TJ, UK
[2]British Trust for Ornithology, The Nunnery, Thetford IP24 2PU, UK
[3]Department of Ornithology, Aranzadi Sciences Society, Zorroagagaina 11, E20014 Donostia, Spain
[4]Department of Zoology and Animal Ecology, Faculty of Biology, University of Latvia, Jelgavas iela 1, Riga, LV-1004, Latvia
[5]Latvian Ornithological Society, Skolas iela 3, Riga, LV-1010, Latvia
[6]Catalan Ornithological Institute, Nat-Museu de Ciències Naturals de Barcelona, Pl. Leonardo da Vinci, 4-5 08019 Barcelona, Spain
[7]Bird Ringing Centre, National Museum, Hornoměcholupská 34, CZ-10200 Praha 10, Czech Republic
[8]Museum and Institute of Zoology, Polish Academy of Sciences, Wilcza 64, 00-679 Warszawa, Poland
[9]Polish Society for the Protection of Birds (OTOP), Odrowaza 24, 05-270 Marki, Poland
[10]Estudio y Seguimiento de Aves SEO/BirdLife, Melquíades Biencinto, Madrid, Spain
[11]Sovon Dutch Centre for Field Ornithology, PO Box 6521, 6503 GA Nijmegen, The Netherlands
[12]Department of Animal Ecology and Physiology, Institute for Water and Wetland Research, Radboud University, PO Box 9010, 6500 GL Nijmegen, The Netherlands
[13]RSPB Centre for Conservation Science, The Lodge, Sandy SG19 2DL, UK
[14]Nord University, Røstad, 7600 Levanger, Norway
[15]BirdLife Norway, Sandgata 30B, 7012 Trondheim, Norway
[16]Centre d'Ecologie et des Sciences de la Conservation (CESCO) UMR 7204, Museum National d'Histoire Naturelle, Paris, France
[17]Norwegian Institute for Nature Research, PO Box 5685 Torgarden, NO-7485 Trondheim, Norway
[18]Finnish Museum of Natural History, FI-00014 University of Helsinki, PO Box 17, Finland
[19]Department of Biology, Lund University, Lund, Sweden
[20]Dansk Ornitologisk Forening, BirdLife Denmark, Vesterbrogade 138-140, DK-1620 København V, Denmark
[21]MME BirdLife Hungary, Monitoring Centre, H-4401 Nyiregyháza 1. PO Box 286, Hungary
[22]SEO/BirdLife, Ciencia Ciudadana, C/Melquiades Biencinto, 34 - 28053 Madrid, Spain
[23]Finnish Museum of Natural History – LUOMUS, PO Box 17, FI-00014, University of Helsinki, Finland
[24]Institute for Environmental Studies, Faculty of Science, Charles University, Prague, Benatska 2, 128 01 Praha 2, Czech Republic
[25]Department of Zoology and Laboratory of Ornithology, Faculty of Science, Palacky University, 771 46 Olomouc, Czech Republic
[26]Czech Society for Ornithology, Na Belidle 34, 150 00 Praha 5, Czech Republic
[27]Swiss Ornithological Institute, Seerose 1, CH-6204 Sempach, Switzerland
[28]Pan-European Common Bird Monitoring Scheme, Czech Society for Ornithology, Na Bělidle, CZ-150 00 Prague 5, Czech Republic
[29]University of Nyíregyháza & MME/BirdLife Hungary, Nyíregyháza, Hungary
[30]BirdLife Österreich, Museumsplatz 1/10/7-8, A-1070 Wien, Austria
[31]Center for Macroecology, Evolution and Climate, Natural History Museum of Denmark, University of Copenhagen, Universitetsparken 15, DK-2100 Copenhagen, Denmark
[32]Swedish Museum of Natural History, Bird Ringing Centre, Box 50007, S-104 05 Stockholm, Sweden

CAM, 0000-0002-4293-2717; RAR, 0000-0003-0504-9906; TC, 0000-0002-9342-2370; FJ, 0000-0002-0606-7332; JAG, 0000-0003-0167-6857

**Author for correspondence:**
Catriona A. Morrison
e-mail: c.morrison@uea.ac.uk

Wildlife conservation policies directed at common and widespread, but declining, species are difficult to design and implement effectively, as multiple environmental changes are likely to contribute to population declines. Conservation actions ultimately aim to influence demographic rates, but targeting actions towards feasible improvements in these is challenging in widespread species with ranges that encompass a wide range of environmental conditions. Across Europe, sharp declines in the abundance of migratory landbirds have driven international calls for action, but actions that could feasibly contribute to population recovery have yet to be identified. Targeted actions to improve conditions on poor-quality sites could be an effective approach, but only if local conditions consistently influence local demography and hence population trends. Using long-term measures of abundance and demography of breeding birds at survey sites across Europe, we show that co-occurring species with differing migration behaviours have similar directions of local population trends and magnitudes of productivity, but not survival rates. Targeted actions to boost local productivity within Europe, alongside large-scale (non-targeted) environmental protection across non-breeding ranges, could therefore help address the urgent need to halt migrant landbird declines. Such demographic routes to recovery are likely to be increasingly needed to address global wildlife declines.

## 1. Background

Across the world, changing climatic conditions and patterns of land use are increasingly driving population declines in species that were previously common and widespread [1]. Efforts to recover widespread but declining populations have typically focussed on identifying and reversing the environmental changes likely to have caused the declines, for example, through the design of agri-environment initiatives that aim to provide key resources in agricultural landscapes [2]. These large-scale, resource-focussed approaches have typically failed to reverse population declines [3], and alternative approaches are urgently needed. Importantly, the actions needed to deliver recovery of a population from a period of decline may not need to address the cause(s) of the decline directly. For example, population declines in several species have been initiated by periods of low survival rates, but recovery has been either facilitated or constrained by subsequent levels of productivity [4,5]. Cases such as these highlight the importance of identifying specific actions capable of influencing demographic rates, and locations in which gains in demographic rate are achievable, rather than relying on generic environmental management approaches in the expectation that this will lead to recovery. Targeting achievable increases in demographic rates could offer new and exciting opportunities to deliver population growth in widespread species of conservation concern, and thus to address the challenges highlighted in the recent Intergovernmental Science-Policy Platform on Biodiversity and Ecosystems Services report [6].

In recent decades, severe population declines in many African-Eurasian migrant landbird species have been reported at both national and international scales across Europe [7–9]. In 2014, parties to the Convention on the Conservation of Migratory Species of Wild Animals adopted the African-Eurasian Migratory Landbirds Action Plan, which is intended to improve the conservation status of migratory landbirds in the region. Recent population declines have been greater in species travelling to the humid tropics of west Africa than those wintering in the arid-zone of sub-Saharan Africa or staying in Europe [7,9–11] (electronic supplementary material, figure S1), but environmental changes anywhere across migratory ranges could be contributing to the declines. While addressing ongoing environmental degradation across Europe and Africa is clearly vital for long-term population persistence, there is an urgent need to implement conservation actions now to slow or halt current migrant declines. Targeting actions to boost specific demographic rates in migratory species could be a fruitful approach to improving the conservation status of these species. For example, efforts to boost productivity might involve the creation of nesting habitat or management of egg or chick predators in locations where productivity is currently low, while efforts to boost survival rates (and perhaps subsequent productivity) might involve the provision of additional food resources in locations and/or time periods when they are scarce. However, such approaches will only be effective if local conditions consistently influence local population trends and in demography and if sites with consistently low demographic rates (survival and/or productivity) can be identified. Regional-scale analyses within the UK have revealed that populations of residents, humid- and arid-zone migrants are all generally faring better in northern than southern regions [12,13], suggesting that opportunities to target actions may exist, but the locations and demographic rate(s) that would need to be targeted have yet to be identified.

Long-term, large-scale surveys of breeding locations across Europe provide data on the extent of spatial variation in abundance and demography, and thus the potential for targeted management of breeding season conditions to influence migrant population declines. As demographic rates can be influenced by the conditions experienced throughout the annual cycle [14], consistent spatial variation in demographic rates of migratory species could reflect effects of local conditions on breeding grounds or effects of conditions experienced elsewhere [15]. However, strong site-level covariation in co-occurring resident and migrant population trends at breeding sites would imply that local breeding season conditions contribute strongly to local population dynamics in both resident and migratory species. In such a case, targeted actions to improve conditions in sites with declining populations could potentially deliver community-wide benefits. By contrast, a lack of site-level covariation in population trends would imply that breeding season conditions alone are not the major driver of local population dynamics in migrants and/or residents, or that the effects of breeding season conditions on migrants and residents differ. In that case, spatial targeting of actions within Europe to improve breeding conditions would be both less achievable (as inconsistent trends would limit identification of suitable sites) and less likely to deliver growth (as local conditions may or may not contribute to local population growth). If site-level covariation in population trends is apparent, strong site-level covariation in levels of either productivity or survival of migrants and residents would identify the rate for which local targeting of conservation actions would be most effective

in delivering local population growth. Consequently, we use citizen-science survey data capturing local abundance and demography of bird species across Europe to quantify the extent and structure of spatial variation and covariation in population trends and demographic rates of co-occurring species with different migratory behaviours.

# 2. Methods

## (a) Abundance metrics from Pan-European common bird monitoring scheme

We used species monitoring data collated under the Pan-European Common Bird Monitoring Scheme (PECBMS: https://pecbms.info/), led by the European Bird Census Council (EBCC), BirdLife International and the Royal Society for the Protection of Birds [16]. In each national scheme, volunteers collect annual count data on the abundance of birds (referred to throughout as abundance) during the breeding season by carrying out either line transects, point counts or territory mapping on survey sites (electronic supplementary material, table S1). We used data from 19 schemes in 17 countries (electronic supplementary material, table S1), covering 13 859 sites and 80 species. We used data collected between 1994 and 2013, with the exact length of time series varying between schemes (electronic supplementary material, table S1). Sites were only included in the analysis if they had been active for three or more years. Species were only included in the analysis if they were present at 15 sites or more.

## (b) Classifying migratory status

Each of the 80 species was classified as either 'resident' (those that stay within Europe during the non-breeding season), 'arid migrant' (species in which the majority of the European population covered by PECBMS winters south of the Sahara, mostly in the arid savannah of the Sahel region) or 'humid migrant' (species in which the majority of the European population covered by the PECBMS winters in the Guinean savannah, humid tropical and other forests south of the Sahel (typified by savannah and forest of west, central, east and southern Africa) (electronic supplementary material, table S2; see [7] for further details of classification).

## (c) Statistical analyses

### (i) Quantifying continent-level population change

In order to confirm previous studies indicating Europe-wide declines in humid-zone migrants and slight increases in the abundance of resident and arid-zone migrant populations [7], we fitted a Gaussian general linear model (GLM) to estimate the average rate of species population change across Europe for each migratory status. In order to account for observer effects, differing sampling protocols and differences in abundance between species (and therefore differences in our capacity to detect changes in abundance), we standardized counts (by subtracting the mean site-level count from the annual count and dividing by the site-level standard deviation) prior to analysis. Annual standardized counts were then modelled as a function of migratory status, year (continuous) and their interaction. All statistical analyses were carried out in R v. 3.1.0 [17].

### (ii) Quantifying site-level population change

For each species at each site, we fitted a GLM to estimate site-level population change. Annual standardized counts were modelled as a function of year (continuous); this year term then describes the relative rate of population change at that site for that species (electronic supplementary material, table S3).

This model resulted in estimates of trends in standardized population abundance (Â) for each species at each site. For simplicity, we use the term 'population trend' hereafter to describe these trends in standardized abundance.

### (iii) Estimating site-level demographic metrics

Data were collated from 10 constant effort site (CES) schemes, spanning eight countries across Europe, all of which use standardized mist-netting during the breeding season to measure the relative productivity and survival of passerine birds [18] (electronic supplementary material, table S4). At each CES, licensed ringers deploy a series of mist-nets in the same positions, for the same length of time, during the morning and/or evening visits, typically between April–May and July–August (the season starts and ends later at higher latitudes). We only included years in which sites were: (i) visited eight or more times in the season (including at least three visits in each of the first and second halves of the season), (ii) had been running for five or more years and, for each species, (iii) on which 25 or more adults and 25 or more juveniles had been captured in total, between 2004 and 2014.

For each species, we estimated site-level mean adult apparent survival rates using the Cormack–Jolly–Seber formulation of mark-recapture models while accounting for transient individuals (electronic supplementary material) and site-level mean productivity as the ratio of the total number of juvenile to adult birds caught at a site during each season, with individuals aged using plumage characteristics (electronic supplementary material, Information). In order to account for differences in species composition between sites, estimates of demographic rates for each species were standardized by subtracting the overall species mean of the site-level estimates and dividing by the site-level standard deviation. This resulted in standardized estimates of survival (Ŝ) and productivity (P̂) for each species at each site.

### (iv) Quantifying site-level mean population trends and demographic rates for resident, arid- and humid-zone migrants

In order to calculate the mean population trend and demographic rate for each migratory status (resident, arid- and humid-zone migrant) at each site, we used a bootstrapping procedure which allowed us to incorporate the error associated with site-level species estimates into the estimates of site-level means for each migratory status category (electronic supplementary material, table S3). For each species at each PECBMS site, we generated 1000 new estimates of population trend ($A_{boot}$) by randomly sampling from a normal distribution with a mean Â and standard deviation $\sigma(Â)$. From these bootstraps, we then calculated 1000 estimates of mean population trend for each migratory status present at each site, taking the mean as the overall site-level estimate and the 97.5th and 2.5th quartiles as the upper and lower confidence limits. This process was repeated for each species at each Euro-CES site, using 1000 new estimates of standardized demographic rate (productivity and survival) generated by randomly sampling from the posterior distribution of Ŝ and P̂ to first generate 1000 estimates of each rate for each species and from these mean site-level estimates of productivity ($P_{boot}$) and survival ($S_{boot}$) for species of each migratory status present at each Euro-CES site.

### (v) Exploring spatial variation in site-level population trends and demographic rates

To explore the variation in mean site-level population trends ($A_{boot}$) and demographic rates ($S_{boot}$, $P_{boot}$) within and between the migratory status categories, we fitted separate Gaussian general linear mixed models (GLMMs) via the R package lme4 [19]. Mean site-level population trends or demographic rates for each migratory status were fitted as the response variable in turn, with migratory status (resident, arid- or humid-zone migrant), latitude

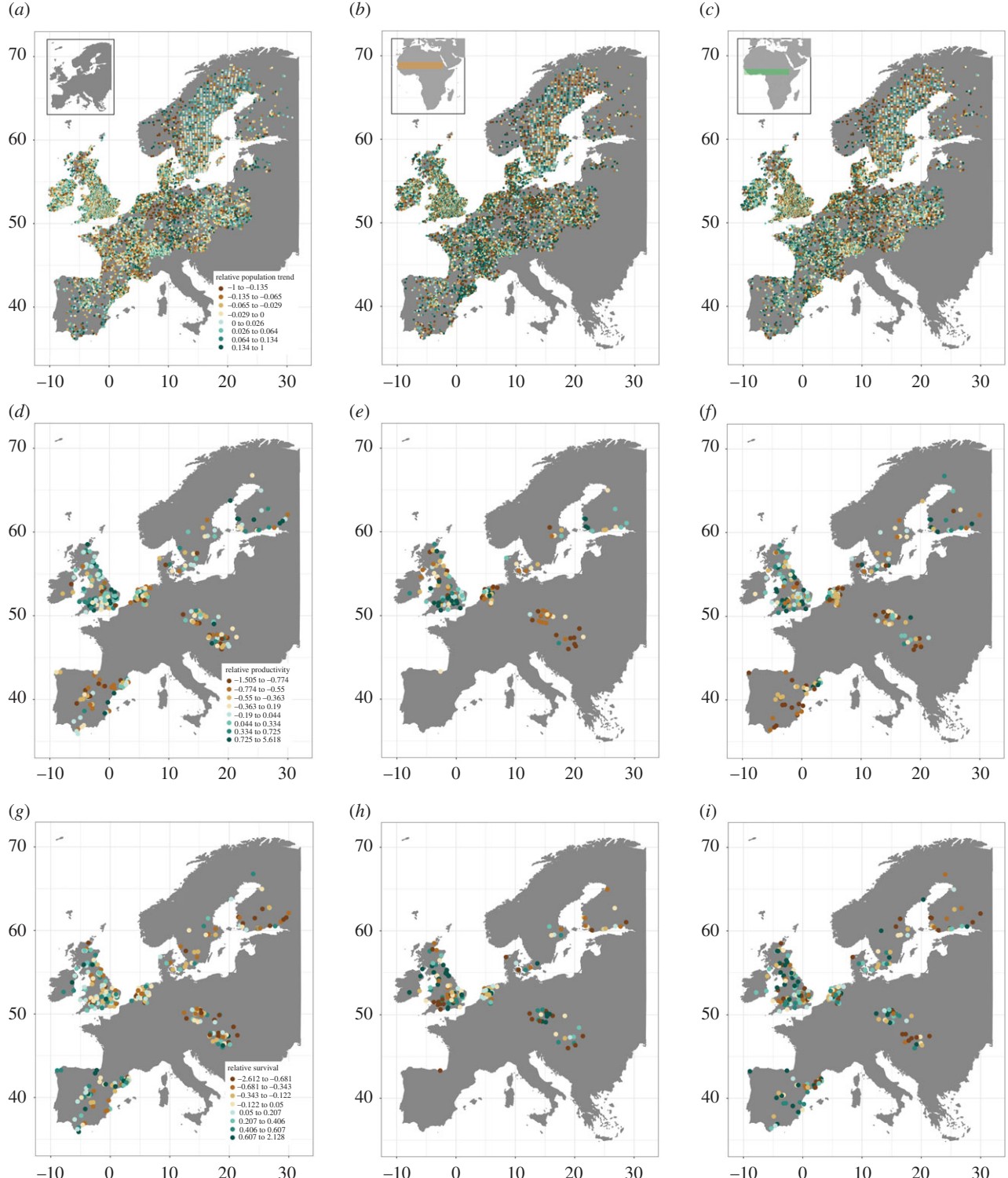

**Figure 1.** Mean site-level trends in abundance between 1994 and 2013 (*a*–*c*), mean standardized site-level productivity between 2004 and 2014 (*d*–*f*), and mean standardized site-level annual survival rates between 2004 and 2014 (*g*–*i*) of resident (*a,d,g*), arid-zone migrant (*b,e,h*), and humid-zone migrant (*c,f,i*) bird species breeding on 13 859 PECBMS sites (*a*–*c*), and 336 Euro-CES sites (*d*–*i*) across Europe. (Online version in colour.)

and longitude, and the interactions between latitude × longitude, migratory status × latitude, and migratory status × longitude as fixed effects. The site was included as a random effect to account for the non-independence of trends from the same sites. To assess the importance of specific effects, we performed a likelihood ratio test by comparing models with and without a particular term, reporting the $\chi^2$ value and associated significance. When interaction terms were found to be significant, the associated main effects were retained in models but we present only the significance of the interaction term and associated parameter estimates. Non-significant interaction terms were removed from the models. We present the results of a final model carried out

on the mean site-level estimates as well as the proportion of times each explanatory variable included in the final model was significant across the 1000 bootstrapped estimates.

### (vi) Quantifying site-level covariation in population trends and demographic rates

Pearson's correlation coefficients were used to estimate the strength of the covariation in mean population trends ($A_{boot}$) and in demographic rates ($S_{boot}$, $P_{boot}$) between residents and each of the two migratory groups (arid-zone and humid-zone). Following [3], for each of our 1000 bootstrapped datasets, we

**Table 1.** Results of GLMMs of the variation in bootstrapped mean site-level (*a*) population trends of resident, arid- and humid-zone migrant bird species breeding at 13 859 PECBMS sites across Europe between 1994 and 2013, (*b*) standardized productivity, and (*c*) standardized adult survival of resident and arid- and humid-zone migrant bird species on 336 Euro-CES sites across Europe between 2004 and 2014, and the proportion of 1000 bootstrapped models reporting significant ($p < 0.05$) effects. (The variance explained by the random effect of site for (*a*) population trends = 0.006 (s.d. = 0.07), (*b*) productivity = 0.26 (s.d. = 0.51), and (*c*) adult survival = 0.04 (0.19). Main effects are included in all models but only presented in the table when interaction terms are not significant (see methods for details).)

| demographic rate | fixed effects | estimate (s.e.) | $\chi^2$ | d.f. | *p*-value | proportion significant ($p < 0.05$) |
|---|---|---|---|---|---|---|
| (*a*) population trend | longitude | −0.0007 (0.0001) | 0.26 | 1 | 0.609 | 0.003 |
| | latitude * migratory status: | | 21.65 | 2 | <0.001 | 1.00 |
| | resident | 0.0003 (0.0003) | | | | |
| | arid | −0.0012 (0.0003) | | | | |
| | humid | −0.0015 (0.0003) | | | | |
| (*b*) productivity | longitude | −0.011 (0.004) | 7.08 | 1 | <0.001 | 0.99 |
| | latitude | 0.041 (0.006) | 39.07 | 1 | <0.001 | 1.00 |
| | migratory status: | | 6.89 | 2 | 0.032 | 0.444 |
| | resident | −2.02 (0.31) | | | | |
| | arid | −2.17 (0.33) | | | | |
| | humid | −2.07 (0.32) | | | | |
| (*c*) adult survival | longitude | −0.014 (0.002) | 33.16 | 1 | <0.001 | 1.00 |
| | latitude | | 0.24 | 1 | 0.628 | 0.006 |
| | migratory status | | 4.16 | 2 | 0.125 | 0.016 |

correlated mean site-level population trend or demographic rate of each migrant group with those of residents and calculated the overall mean correlation coefficient and the 97.5th and 2.5th quantile of the distribution of the correlation coefficients as the upper and lower confidence intervals. Significant associations were identified as those in which the 97.5th and 2.5th quantiles did not overlap zero.

To estimate the mean difference in site-level population trends or demographic rates of residents and each of the two migratory groups (arid-zone and humid-zone), we calculated the mean difference (migrant—resident at each site) for each of our 1000 bootstrapped datasets. Significant differences were identified as those in which the 97.5th and 2.5th quantiles did not overlap zero.

To explore the effects of spatial autocorrelation on these patterns this process was repeated within each scheme and the results presented in the electronic supplementary material, tables S8–S10 and figures S3–S8.

## 3. Results

### (a) European population trends and migratory strategy

Across the 13 859 European survey sites, overall mean population trends between 1994 and 2013 were similar and slightly positive for residents and arid-zone migratory species, but humid-zone species declined significantly (electronic supplementary material, figure S1 and table S5).

### (b) Site-level variation in population trends and demography

Across 13 859 PECBMS sites, mean population trends of resident (46 species), arid-zone migrant (15 species) and humid-zone migrant (19 species) species varied greatly between sites, with local declines and increases occurring in all three groups across all 17 countries (figure 1*a–c*). No strong

geographical structure in mean site-level population trends was apparent in any group (figure 1*a–c*), although populations in the east and north of Europe tended to be faring slightly less well on average (table 1). Across 336 Euro-CES sites at which demography was monitored, mean standardized productivity and survival of resident (18 species), arid-zone migrants (three species) and humid-zone migrants (five species) also varied greatly (figure 1*d–i*). Again, no strong geographical structuring of demography was evident, although productivity tended to be slightly lower in the east and south, while survival rates were slightly lower in the east (figure 1 and table 1). Thus, high levels of local variation are apparent in population trends and demography of these species, and there is little evidence of large-scale clustering of sites with similar trends in abundance or mean levels of demography.

### (c) Site-level covariation in population trends

Mean site-level population trends of both arid- and humid-zone migrant species covaried positively and significantly with population trends of co-occurring resident species, with the strongest association between resident and humid-zone species (figure 2*a,b* and table 2). The slope of the covariation differs significantly from unity (table 2) and migrants tend to be faring less well than residents at sites with increasing population trends (figure 2*a,b*, upper right quadrant) while, at sites with population declines, migrants tend to be faring slightly better than residents (figure 2*a,b*, lower left quadrant).

Humid-zone migrants are the only group of species declining overall [7] (electronic supplementary material, figure S1) and site-level trends of humid-zone migrants were significantly lower than those of co-occurring resident

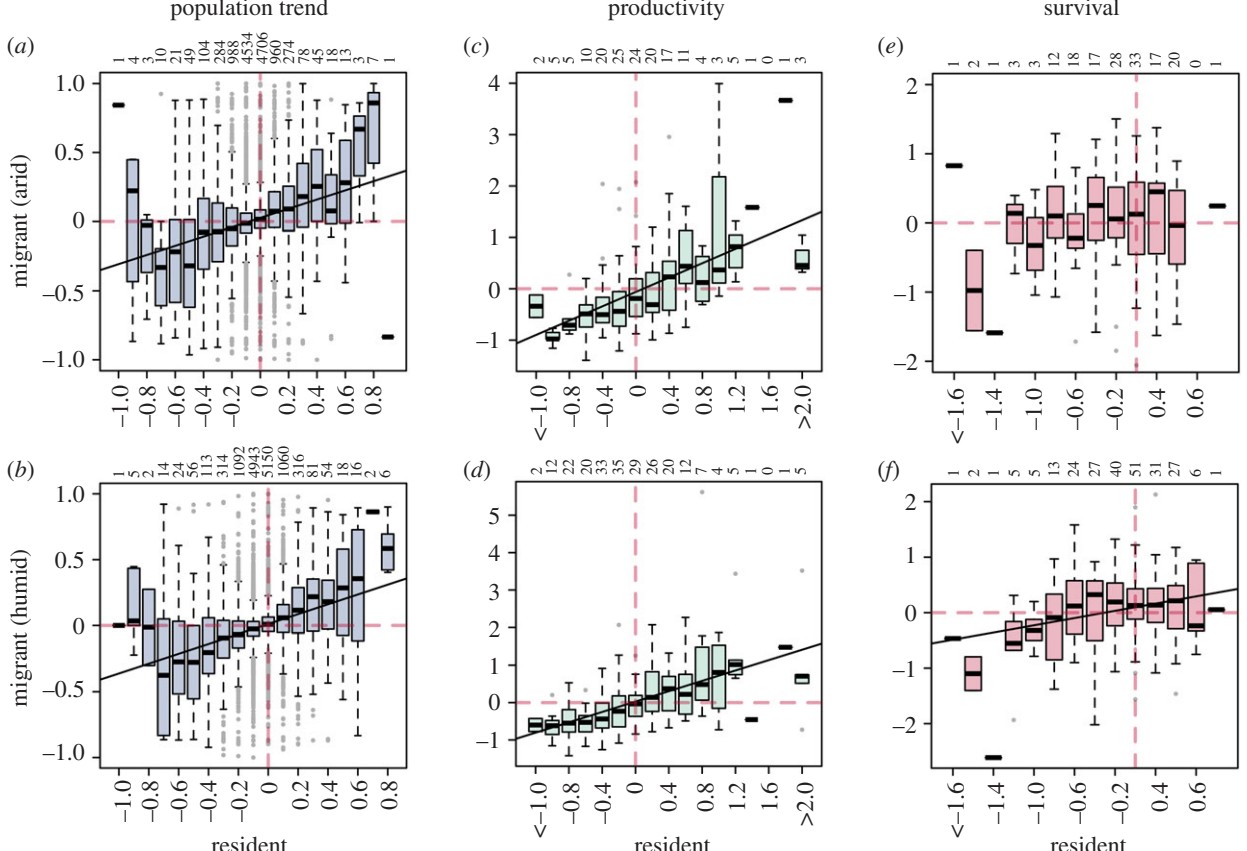

**Figure 2.** Covariation between resident bird species and their co-occurring arid-zone (*a,c,e*), and humid-zone (*b,d,f*) migrant species in mean site-level (*a,b*) population trends (*a*: 12 103 sites; *b*: 13 267 sites), (*c,d*) standardized mean site-level productivity (*c*: 156 sites; *d*: 247 sites), and (*e,f*) standardized mean site-level annual survival rates (*e*: 156 sites; *f*: 247 sites). Lines of best fit are shown for significant associations and numbers indicate the number of sites. Horizontal bars indicate medians, boxes indicate interquartile range, whiskers indicate minimum and maximum values and circles indicate values 1.5 times higher or lower than 1st and 3rd interquartile, respectively. (Online version in colour.)

species (table 2). Interestingly, while there is no overall significant difference between the population trends of arid-zone migrants and residents (electronic supplementary material, figure S1), site-level population trends of arid-zone migrants were significantly higher than those of co-occurring resident species (table 2). This disparity suggests possible differences in distribution, with arid-zone species disproportionately occurring in sites with either no residents and/or not occurring in sites where residents are doing well. These patterns were apparent even when models were restricted to sites that had been surveyed for seven or more years (electronic supplementary material, table S7). These patterns were also apparent within survey schemes, suggesting that they are consistent across Europe (electronic supplementary material, table S8 and figures S3 and S4).

### (d) Site-level covariation in demography
Covariation in the demographic rates of resident and migrant species was also apparent, with mean site-level productivity of resident species showing much stronger covariation with that of both arid- and humid-zone migrants (figure 2*c,d* and table 2) than in equivalent mean site-level survival rates (figure 2*e,f* and table 2). The marginally significant covariation in survival rates of residents and humid-zone migrants was not present when models were restricted to sites that had been surveyed for seven or more years (electronic supplementary material, table S7). As with covariation in population trends, these patterns were also

apparent within survey schemes (electronic supplementary material, tables S9 and S10, figures S5–S8).

## 4. Discussion
Our site-level trend analyses reveal covariation in local population trends of migrants and residents, such that co-occurring species tend to have similar directions and magnitudes of change. Consequently, sites that are good for resident species tend to be good for migrants, and *vice versa*. This suggests that local breeding season conditions are a realistic target for conservation actions which should be effective across the avian community. Similarly positive, migrant-resident covariation in productivity, but not survival, suggests that actions targeted at boosting local productivity within Europe have the potential to benefit local populations of both migrant and resident species.

Concerns over the potential contribution of environmental changes within African humid-zone wintering grounds to migrant population trends (through impacts on annual survival probabilities) have arisen because of the concentration of declines among species travelling to these areas [7,9]. However, while greater overall population declines in humid-zone migrants could be viewed as evidence for current 'costs of being migratory', the demographic rates that underpin these declines can be influenced by processes operating anywhere within their geographical ranges and across the annual cycle. For example, humid-zone migrants could

**Table 2.** Results of bootstrapped Pearson correlations of associations, differences and regression coefficients between mean site-level population trends and demographic rates of resident bird species and co-occurring migratory bird species of differing status (arid-zone and humid-zone) on 13 859 PECBMS survey sites and 336 Euro-CE sites across Europe. (*Indicates significant differences from zero (or from unity, in the case of regression coefficients).)

| demographic rate | migratory status | mean correlation coefficient (95% CIs) | mean difference migrant – resident (95% CIs) | mean regression coefficient (95% CIs) |
|---|---|---|---|---|
| population change | arid | 0.12 (0.10–0.15)* | 0.010 (0.005 to 0.013)* | 0.26 (0.21–0.32)* |
| | humid | 0.18 (0.15–0.20)* | −0.007 (−0.010 to −0.004)* | 0.30 (0.25–0.34)* |
| productivity | arid | 0.44 (0.35–0.52)* | −0.17 (−0.20 to −0.15)* | 0.60 (0.46–0.71)* |
| | humid | 0.48 (0.42–0.53)* | −0.06 (−0.08 to −0.04)* | 0.60 (0.51–0.69)* |
| adult survival | arid | 0.06 (−0.08–0.21)$^{ns}$ | 0.14 (0.08 to 0.20)* | 0.09 (−0.12–0.35)* |
| | humid | 0.14 (0.03–0.26)* | 0.12 (0.07 to 0.16)* | 0.19 (0.03–0.35)* |

be experiencing greater risks of harsh environmental conditions on their migratory journeys [20], while their later arrival on breeding grounds could mean that they are less able to cope with changing breeding conditions [21] or, should nest loss rates be high, they may lack the time to lay replacement clutches [22]. Furthermore, weak migratory connectivity is typical of many species [23,24], with individuals from the same breeding population often separated by hundreds or thousands of kilometres on their wintering grounds. Consequently, although efforts to maintain important habitats across Africa will clearly be crucial to the long-term conservation of both African-Eurasian migrants and African resident species, delivering population recovery for species in particular parts of their breeding range by targeting actions at locations within Africa is unlikely to be achievable. By contrast, the strong natal and breeding site fidelity that is typical of migratory bird species [25] suggests that delivering population recovery through actions targeted on breeding grounds will be more feasible.

Importantly, the demographic factors that lead to population decline are not necessarily the factors that can be most easily influenced to reverse those declines [4,26]. The weak covariation in site-level adult annual survival rates of migrant and resident species suggests they are influenced by conditions experienced throughout the annual cycle, with survival rates measured on breeding grounds integrating the effects of conditions experienced by individuals across their migratory range (e.g. droughts in the arid-zone [27], storms during the migratory journey [28]). Designing specific conservation actions to boost annual survival rates would therefore be highly challenging. By contrast, the strong covariation in productivity of migrants and residents demonstrated by Euro-CES data provides a route for identifying the conditions associated with high and low levels of productivity, and manipulating local environments to increase the frequency of sites achieving high productivity. For example, low productivity can be particularly prevalent in fragmented landscapes, when small, isolated populations fail to attract sufficient females [29,30], or in areas that are intensively managed [31]. Consequently, targeting resources to increase the size and quality of breeding habitats in fragmented landscapes could be an effective tool for increasing the frequency of high productivity sites, particularly as relevant resources and infrastructure exist through European agri-environment schemes [2] and protected area networks

[32] in contrast with much of sub-Saharan Africa. The actions needed to deliver on international agreements to improve the conservation status of migratory landbirds are therefore likely to comprise targeted local improvements of breeding conditions across Europe, alongside large-scale (non-targeted) environmental protection of key habitats across non-breeding ranges.

## 5. Conclusion

Rapid declines in widespread species are occurring throughout the world, and there is an urgent need to identify actions capable of addressing these declines. Citizen-science data hold unique information that can be used to connect large-scale patterns with local-scale processes to target and design conservation actions on the ground. Exploiting these data to identify consistent spatial variation in population trends and, especially, demography can be an extremely useful tool in diagnosing the most fruitful targets for interventions. These findings suggest an approach of targeted actions to boost local productivity within Europe, alongside large-scale (non-targeted) environmental protection across non-breeding ranges, may provide the best hope for halting, and perhaps even reversing, the rapid population declines in humid-zone migrants and potentially other species as well.

Data accessibility. The datasets supporting this article can be obtained from Dryad Digital Repository: https://doi.org/10.5061/dryad.76hdr7svs [33].

Authors' contributions. C.A.M., S.J.B., R.A.R., J.A.C. and J.A.G. conceived and wrote the study; C.A.M. performed the analysis and all other authors provided the data and commented on the manuscript.

Competing interests. We declare we have no competing interests.

Funding. This study was funded by NERC (project NE/L007665/1 and NE/T007/354/1). A.Le. was funded by the Academy of Finland (grant no. 275606), J.R. by project PRIMUS/17/SCI/16 and J.C. by the Ministry of Culture of the Czech Republic (DKRVO 2018/15, National Museum, 00023272).

Acknowledgements. This study was made possible by strong Pan-European collaborations and friendships and is the result of thousands of hours of fieldwork by dedicated volunteers. We thank all the volunteers of the national monitoring and ringing schemes; the Latvian Nature Conservation Agency, Spanish Ornithological Society (SEO/BirdLife), Norwegian Climate and Environment Ministry, Norwegian Environment Agency, Swedish Environmental Protection Agency, Centre for Animal Movement Research (CAnMove) and Biodiversity and Ecosystem Services in a Changing Climate (BECC), Austrian Ministry for Sustainability and Tourism, the RSPB and JNCC (on behalf of

the UK Statutory Nature Conservation Bodies) for support of national schemes; Zdeněk Vermouzek and Petr Voříšek for scheme coordination, Juliet Vickery for contributions to study development, and three anonymous reviewers for helpful comments on the manuscript.

Much of the research presented in this paper was carried out on the High Performance Computing Cluster supported by the Research and Specialist Computing Support service at the University of East Anglia.

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
