## [Peer Review File · Proceedings of the Royal Society B: Biological Sciences]

Review History

RSPB-2020-2955.R0 (Original submission)

Review form: Reviewer 1

Recommendation

Accept with minor revision (please list in comments)

Scientific importance: Is the manuscript an original and important contribution to its field?

Excellent

General interest: Is the paper of sufficient general interest?

Excellent

Quality of the paper: Is the overall quality of the paper suitable?

Excellent

Is the length of the paper justified?

Yes

Should the paper be seen by a specialist statistical reviewer?

No

Do you have any concerns about statistical analyses in this paper? If so, please specify them explicitly in your report.

No

It is a condition of publication that authors make their supporting data, code and materials available - either as supplementary material or hosted in an external repository. Please rate, if applicable, the supporting data on the following criteria.

Is it accessible?

Yes

Is it clear?

Yes

Is it adequate?

Yes

Do you have any ethical concerns with this paper?

No

Comments to the Author

The authors examine large-scale patterns of population trends, productivity and survival across breeding birds in Europe and test the hypothesis that co-occurring species with different migration behaviour share similar population trends and demographic rates. The results show significant covariation in local population trends and productivity, but not survival, of migrants and residents.

The original version of the manuscript was examined by two reviewers, who all saw great potential in the paper but also had concerns, particularly in relation to the methodology and the clarity of some parts of the introduction and methods sections.

In my opinion, the authors have properly addressed all of the concerns raised by the editor and previous reviewers. I have found the revised version of the manuscript well written and clear to follow.

I only have few minor comments:

Lines 309-314: The beginning of the discussion section contains information that is already partially provided in the introduction section. I understand that this introductory paragraph may be useful to remind the reader of the background to this study but, given the page constraints, I suggest that the authors check whether this is really needed or if some space could be made, for example to explicitly acknowledge the limitations of the study or to incorporate some of the information that is currently shown in the supplementary materials.

Lines 316 and 320: 'strong' covariation: I think this is a bit overstated here. I am aware that we cannot expect very high correlation coefficients in such complex systems, however $r=0.12$ or $r=0.18$ mean that only a small proportion of the variance is shared. I wonder if a more 'neutral' wording ('significant' covariation) would be more appropriate?

Line 507: 'circles' is misspelled as 'cicles'

Supplementary information: the first paragraph ("European population trends and migratory strategy") contains a fairly important result. Could this be integrated into the main text? I understand that there are page constraints, so it may not be possible, but perhaps some space could be made by condensing the beginning of the discussion section?

Supplementary information, lines 74-76: Does this not make 1200 parameter estimates rather than 1000?

Decision letter (RSPB-2020-2955.R0)

15-Jan-2021

Dear Dr Morrison:

Your manuscript has now been peer reviewed and the reviews have been assessed by an Associate Editor. The reviewers' comments (not including confidential comments to the Editor) and the comments from the Associate Editor are included at the end of this email for your reference. As you will see, the reviewers and the Editors have raised some concerns with your manuscript and we would like to invite you to revise your manuscript to address them.

Research ethics:

Use of animals and field studies:

It is a condition of publication that you make available the data and research materials supporting the results in the article. Please see our Data Sharing Policies (<https://royalsociety.org/journals/authors/author-guidelines/#data>). Datasets should be deposited in an appropriate publicly available repository and details of the associated accession number, link or DOI to the datasets must be included in the Data Accessibility section of the

article (<https://royalsociety.org/journals/ethics-policies/data-sharing-mining/>). Reference(s) to datasets should also be included in the reference list of the article with DOIs (where available).

[http://datadryad.org/submit?journalID=RSPB&manu=\(Document not available\)](http://datadryad.org/submit?journalID=RSPB&manu=(Document%20not%20available)), which will take you to your unique entry in the Dryad repository.

Please submit a copy of your revised paper within three weeks. If we do not hear from you within this time your manuscript will be rejected. If you are unable to meet this deadline please let us know as soon as possible, as we may be able to grant a short extension.

Best wishes,
Dr Sasha Dall
mailto: proceedingsb@royalsociety.org

Associate Editor
Board Member: 1
Comments to Author:

I thank the authors for their extensive revisions to the manuscript. I believe this has much improved in clarity and appeal to a general! The introduction is now much easier to follow and clearly sets out the ideas in the paper. The methods and results are succinct in the main text, and the new analyses test for spatial heterogeneity and propagate the uncertainty in abundance, productivity, and survival estimates as was suggested in the last round of peer review. The new figures are really interesting and convey the main results very well. The supplement gives vital supporting analyses that convinced me of the validity of conclusions drawn in the main text. Finally, the discussion puts these results nicely into a bigger context.

The review we obtained is also very positive (unfortunately, we were unable to get reviews by the two original reviewers at this time). The new reviewer has some small final suggestions that I

agree would improve the study further, as most of these were points I was also wondering about; in particular including that one sentence about population trends and migratory strategy (currently start of supplementary information) in the main text. In addition, one tiny comment from me: Supplementary table 10 is missing the species numbers.

Reviewer(s)' Comments to Author:

Referee: 1

Comments to the Author(s)

The authors examine large-scale patterns of population trends, productivity and survival across breeding birds in Europe and test the hypothesis that co-occurring species with different migration behaviour share similar population trends and demographic rates. The results show significant covariation in local population trends and productivity, but not survival, of migrants and residents.

The original version of the manuscript was examined by two reviewers, who all saw great potential in the paper but also had concerns, particularly in relation to the methodology and the clarity of some parts of the introduction and methods sections.

In my opinion, the authors have properly addressed all of the concerns raised by the editor and previous reviewers. I have found the revised version of the manuscript well written and clear to follow.

I only have few minor comments:

Lines 309-314: The beginning of the discussion section contains information that is already partially provided in the introduction section. I understand that this introductory paragraph may be useful to remind the reader of the background to this study but, given the page constraints, I suggest that the authors check whether this is really needed or if some space could be made, for example to explicitly acknowledge the limitations of the study or to incorporate some of the information that is currently shown in the supplementary materials.

Lines 316 and 320: 'strong' covariation: I think this is a bit overstated here. I am aware that we cannot expect very high correlation coefficients in such complex systems, however $r=0.12$ or $r=0.18$ mean that only a small proportion of the variance is shared. I wonder if a more 'neutral' wording ('significant' covariation) would be more appropriate?

Line 507: 'circles' is misspelled as 'cicles'

Supplementary information: the first paragraph ("European population trends and migratory strategy") contains a fairly important result. Could this be integrated into the main text? I understand that there are page constraints, so it may not be possible, but perhaps some space could be made by condensing the beginning of the discussion section?

Supplementary information, lines 74-76: Does this not make 1200 parameter estimates rather than 1000?

Author's Response to Decision Letter for (RSPB-2020-2955.R0)

See Appendix A.

Decision letter (RSPB-2020-2955.R1)

05-Feb-2021

Dear Dr Morrison

I am pleased to inform you that your manuscript entitled "Covariation in population trends and demography reveals targets for conservation action." has been accepted for publication in Proceedings B.

Open Access

Your article has been estimated as being 8 pages long. Our Production Office will be able to confirm the exact length at proof stage.

Paper charges

Sincerely,

Dr Sasha Dall

Appendix A

Dear Dr Dall,

Many thanks for your supportive comments and for our manuscript ID RSPB 2019 2465. We have addressed all of the remaining comments, and below we describe our responses to each point (in bold).

Yours sincerely,
Catriona Morrison & co-authors

Associate Editor
Board Member: 1
Comments to Author:

I thank the authors for their extensive revisions to the manuscript. I believe this has much improved in clarity and appeal to a general! The introduction is now much easier to follow and clearly sets out the ideas in the paper. The methods and results are succinct in the main text, and the new analyses test for spatial heterogeneity and propagate the uncertainty in abundance, productivity, and survival estimates as was suggested in the last round of peer review. The new figures are really interesting and convey the main results very well. The supplement gives vital supporting analyses that convinced me of the validity of conclusions drawn in the main text. Finally, the discussion puts these results nicely into a bigger context.

The review we obtained is also very positive (unfortunately, we were unable to get reviews by the two original reviewers at this time). The new reviewer has some small final suggestions that I agree would improve the study further, as most of these were points I was also wondering about; in particular including that one sentence about population trends and migratory strategy (currently start of supplementary information) in the main text.

As suggested we have moved this section from the SOM to the main results section.

In addition, one tiny comment from me: Supplementary table 10 is missing the species numbers.

These numbers are the number of sites and have been added to Supplementary table 10.

Reviewer(s)' Comments to Author:

Referee: 1

Comments to the Author(s)

The authors examine large-scale patterns of population trends, productivity and survival across breeding birds in Europe and test the hypothesis that co-occurring species with different migration behaviour share similar population trends and demographic rates. The results show significant covariation in local population trends and productivity, but not survival, of migrants and residents. The original version of the manuscript was examined by two reviewers, who all saw great potential in the paper but also had concerns, particularly in relation to the methodology and the clarity of some parts of the introduction and methods sections.

In my opinion, the authors have properly addressed all of the concerns raised by the editor and previous reviewers. I have found the revised version of the manuscript well written and clear to follow.

I only have few minor comments:

Lines 309-314: The beginning of the discussion section contains information that is already partially

provided in the introduction section. I understand that this introductory paragraph may be useful to remind the reader of the background to this study but, given the page constraints, I suggest that the authors check whether this is really needed or if some space could be made, for example to explicitly acknowledge the limitations of the study or to incorporate some of the information that is currently shown in the supplementary materials.

We have now edited this paragraph to make it shorter, giving space to add the information from the supplementary material as suggested.

Lines 316 and 320: 'strong' covariation: I think this is a bit overstated here. I am aware that we cannot expect very high correlation coefficients in such complex systems, however $r=0.12$ or $r=0.18$ mean that only a small proportion of the variance is shared. I wonder if a more 'neutral' wording ('significant' covariation) would be more appropriate?

We have removed the word 'strong' in both instances.

Line 507: 'circles' is misspelled as 'cicles'

This has been corrected.

Supplementary information: the first paragraph ("European population trends and migratory strategy") contains a fairly important result. Could this be integrated into the main text? I understand that there are page constraints, so it may not be possible, but perhaps some space could be made by condensing the beginning of the discussion section?

As suggested we have moved this section from the SOM to the main results section and condensed the start of the discussion to allow room.

Supplementary information, lines 74-76: Does this not make 1200 parameter estimates rather than 1000?

This was a mistake, we have change it to 1000.